# 2-Butyl-2-tert-butyl-5,5-diethylpyrrolidine-1-oxyls: Synthesis and Properties

**DOI:** 10.3390/molecules25040845

**Published:** 2020-02-14

**Authors:** Irina F. Zhurko, Sergey Dobrynin, Artem A. Gorodetskii, Yuri I. Glazachev, Tatyana V. Rybalova, Elena I. Chernyak, Nargiz Asanbaeva, Elena G. Bagryanskaya, Igor A. Kirilyuk

**Affiliations:** 1Novosibirsk Institute of Organic Chemistry, Siberian Branch of Russian Academy of Sciences, Academician Lavrentiev Ave. 9, Novosibirsk 630090, Russia; zhurko@nioch.nsc.ru (I.F.Z.); s.a.dobrynin@gmail.com (S.D.); gorodaa@nioch.nsc.ru (A.A.G.); rybalova@nioch.nsc.ru (T.V.R.); chernyak@nioch.nsc.ru (E.I.C.); nargiz-asan@mail.ru (N.A.); 2Institute of Chemical Kinetics & Combustion SB RAS, Institutskaya 3, Novosibirsk 630090, Russia; glaza@ngs.ru; 3Novosibirsk State University, 2 Pirogova St., Novosibirsk 630090, Russia

**Keywords:** nitroxides, nitrones, 1,3-dipolar cycloaddition reaction, organolithium compounds, EPR, spin probes, reduction kinetics

## Abstract

Nitroxides are broadly used as molecular probes and labels in biophysics, structural biology, and biomedical research. Resistance of a nitroxide group bearing an unpaired electron to chemical reduction with low-molecular-weight antioxidants and enzymatic systems is of critical importance for these applications. The redox properties of nitroxides are known to depend on the ring size (for cyclic nitroxides) and electronic and steric effects of the substituents. Here, two highly strained nitroxides, 5-(*tert*-butyl)-5-butyl-2,2-diethyl-3-hydroxypyrrolidin-1-oxyl (**4**) and 2-(*tert*-butyl)-2-butyl-5,5-diethyl-3,4-bis(hydroxymethyl)pyrrolidin-1-oxyl (**5**), were prepared via a reaction of the corresponding 2-*tert*-butyl-1-pyrroline 1-oxides with butyllithium. Thermal stability and kinetics of reduction of the new nitroxides by ascorbic acid were studied. Nitroxide **5** showed the highest resistance to reduction.

## 1. Introduction

Nitroxides are broadly used as molecular probes and labels in biophysics, structural biology, and biomedical research [1,2,3,4,5,6]. Modern trends in these research fields, such as *in-cell* electron paramagnetic resonance (EPR)/pulsed electron-electron double resonance (PELDOR) experiments [7,8] and in vivo NMR and EPR imaging [9,10], require stable paramagnetic agents that can retain the radical center in live cells or tissues long enough for the measurements. Resistance of a nitroxide group bearing an unpaired electron to chemical reduction with low-molecular-weight antioxidants and enzymatic systems is crucial for these applications. The redox properties of nitroxides depend on the ring size (for cyclic nitroxides) and electronic and steric effects of the substituents [11,12]. Five-membered-ring nitroxides of the pyrrolidine series show the highest resistance to reduction [13,14]. Introduction of bulkier alkyl substituents instead of methyls at the carbon atom adjacent to the N–O**⋅** group can increase nitroxide stability in model systems and in biological samples [15,16,17]. 2,2,5,5-Tetraethylpyrrolidine-1-oxyls **1** and **2** are the most resistant to reduction among all currently known nitroxides ([Fig molecules-25-00845-ch001]) [18,19,20]. One can assume that a further increase in the steric demand of alkyls at positions 2 and 5 can make a nitroxide even more stable, but the synthesis of such strained structures remains a challenge. We have previously published efficient synthesis of a spirocyclic nitroxide of the 2,5-dihydroimidazole series, **3**, with *tert*-butyl and *n*-butyl substituents at position 5 of the imidazole ring via the reaction of corresponding cyclic α-*tert*-butylnitrone with *n*-butyllithium (*n*-BuLi). Regretfully, this nitroxide appears to be thermally unstable, rapidly decomposing at 80 °C via *tert*-butyl radical abstraction [21]. Here we used the same strategy to synthesize highly strained nitroxides of the pyrrolidine series. Two nitroxides, **4** and **5**, were prepared via a reaction of corresponding 2-*tert*-butyl-1-pyrroline-1-oxides with BuLi. Thermal stability and redox properties of the new nitroxides were studied. The new nitroxides are very lipophilic compounds with poor solubility in water; however, their hydroxyl groups may be employed for chemical binding to a delivery vehicle. The utility of lipophilic nitroxides with large hydrophobic groups for the design of lipid nanoparticles or micelles for MRI contrast and MRI-controlled drug delivery has been demonstrated by various research groups [22,23,24].

## 2. Results

### 2.1. Synthesis of Nitroxide 4

The new nitroxide, **4**, was synthesized from 5-(tert-butyl)-2,2-diethyl-3-oxo-3,4-dihydro-2H-pyrrole 1-oxide (**6**) (Scheme 1).

The general procedure for the synthesis of similar 1-pyrroline 1-oxides has been developed by Reznikov et al. [25]. Using a procedure similar to the one described in the literature [26], 1-hydroxy-5,5-diethyl-2,2,4-trimethyl-2,5-dihydroimidazole **7** [27] was treated with an excess of lithium diisopropylamide to produce dianion intermediate **8**, which reacts with ethyl pivalate to form **9** (Scheme 2). Quenching of the reaction mixture in aerobic conditions leads to partial oxidation into the corresponding nitroxide; therefore, the mixture was treated with manganese dioxide to finalize the conversion. Nitroxide **10** was isolated as an orange paramagnetic crystalline solid with spectral parameters close to those of known 4-(2-oxoalkylidene)-imidazolidines: strong absorption at 298 nm in the UV spectrum indicated formation of an enaminoketone conjugated system, IR-spectra of **10** showed characteristic strong vibration bands of the enaminoketone moiety at 1635 and 1561 cm^−1^ (cf. [26]). Hydrogenation of **10** on a palladium catalyst afforded hydroxylamine **9**. The structure of **9** was confirmed by a ^1^H NMR spectrum: a methine proton signal at 5.01 ppm and a broad line of NH at 5.35 ppm indicate that **9** predominantly exists in 4-(2-oxoalkylidene)-imidazolidine form in CDCl_3_.

Treatment of enaminoketone **9** with conc. hydrochloric acid for 82 h resulted in hydrolysis of the heterocycle, and subsequent neutralization of the reaction mixture to pH 7–8 led to recyclization with the formation of **6** (Scheme 2). IR, UV, and NMR spectra of **6** are close to those provided in the literature for similar compounds [25]. According to NMR data, **11** exists in CDCl_3_ as a mixture of tautomeric forms (**6A** and **6B**) in a 4:7 ratio (Scheme 2) (cf. [25]). The presence of the N-hydroxyenamine form **6B** makes 3-oxo-3,4-dihydro-2*H*-pyrrole 1-oxides susceptible to oxidation to vinylnitroxide radicals, which in turn are prone to dimerization [28]. Indeed, similarly to other known 3-oxo-3,4-dihydro-2*H*-pyrrole 1-oxides, **6** undergoes oxidation by air oxygen in solutions, thus yielding the emergence of deep violet coloring due to ethylene-type dimer formation (Scheme 3). Complete conversion to dimer **12** can be achieved via oxidation of **6** with manganese dioxide. The samples of similar dimers are known to have quintet EPR spectra, initially attributed to a contribution of the biradical form [28]. It has been demonstrated later that the signal belongs to paramagnetic impurities [29], presumably to radical ions of the dinitrones [30]. Exchange with these radical ions may account for the strong broadening of some signals in ^13^C NMR spectra of **12** (see Appendix A). Single-crystal X-ray analysis of **12** uncovered a dihedral angle of 60° between nearly planar pyrrolinone rings (Figure 1; deviation within ±0.032 Å) (cf. [29]).

Of note, earlier processing of the reaction mixture in the hydrolysis of **9** leads to the formation of byproducts and decreases the yield of **6**. If the reaction mixture was processed, after 40 h, another compound (**13**, yield 15%) was isolated along with **6** (Scheme 2). Obviously, enaminoketone **9** hydrolysis proceeds via the consecutive formation of acyclic enaminoketone **14** and β-diketone **15** intermediates, both present in the reaction mixture, and formation of **13** is a result of cyclization of the former. Formation of similar compounds has been previously observed in a recyclization reaction of imidazolidine enaminoketones with an acceptor group at the exo-methylene carbon atom [31].

According to NMR data, in CDCl_3_, **13** exists predominantly as tautomer **13A**. Enaminonitrone **13** is unstable in aerobic conditions at ambient temperature and undergoes oxidation to 5-(*tert*-butyl)-2,2-diethyl-3-imino-4-oxo-3,4-dihydro-2*H*-pyrrole 1-oxide (**16**), which is then hydrolyzed by air moisture into 5-(*tert*-butyl)-2,2-diethyl-3,4-dioxo-3,4-dihydro-2*H*-pyrrole 1-oxide (**17**) (Scheme 3). Formation of similar α-diketones and their monoimines has been observed previously during studies on the reactions of 4-chloro-2,2-dimethyl-3-oxo-5-phenyl-3,4-dihydro-2*H*-pyrrole 1-oxide [32,33].

It is known that treatment of 3-oxo-3,4-dihydro-2*H*-pyrrole 1-oxides with organometallic reagents leads to the formation of enolate salts, which nevertheless remain capable of reacting with the second equivalent of a Grignard reagent or an organolithium compound via nucleophilic addition onto a nitrone carbon. A number of pyrrolidine nitroxides have been prepared via this reaction, including 5-(*tert*-butyl)-2,2-dimethyl-1-oxo-5-phenylpyrrolidin-1-oxyl [34]. It is noteworthy that the addition of EtMgBr to **6** did not occur either at ambient temperature or upon heating. After heating to reflux in THF with a large excess of EtMgBr for 15 h, the nitrone was almost quantitatively recovered from the reaction mixture. In contrast, **6** readily reacted with BuLi within 1.5 h, and quenching of the reaction mixture with water under aerobic conditions afforded nitroxide **18** with a 60% yield (Scheme 3).

Nitroxide **18** is unstable, slowly decomposing at room temperature or upon heating, with deep crimson coloring typical for 3-oxo-3,4-dihydro-2*H*-pyrrole 1-oxide dimers. We already mentioned above the thermal instability of nitroxide **3** [21]. Here, we followed a similar approach to investigate thermal decomposition of **18**: a mixture of **18** with a 10-fold excess of 2,2,6,6-tetramethylpiperidine-1-oxyl (**19**, TEMPO) was heated to 80 °C under anaerobic conditions. The conversion was complete within 7 days, affording alkoxyamine **20** and dimer **21** (Scheme 4). Obviously, TEMPO is acting both as a spin trap for the *tert*-butyl radical and as an oxidant converting in situ–formed 3-oxo-3,4-dihydro-2*H*-pyrrole 1-oxide **22** into dimer **21**.

Reduction of **18** with sodium borohydride gave nitroxide **4** as a single diastereomer (racemic mixture) according to HPLC (Appendix A). Single-crystal X-ray analysis (Figure 1) revealed that the hydroxyl group is at cis-position to the *tert*-butyl group, apparently indicating kinetic control of the reduction reaction in which the reagent approaches the C=O bond from the less hindered side.

### 2.2. Synthesis of Nitroxide **5**

This synthesis is based on a recently developed procedure for assembling sterically hindered pyrrolidines via a three-component domino process [20]. A mixture of *tert*-leucine, dimethyl fumarate, 2-pentanone, and DMF in toluene was heated to reflux with a Dean–Stark apparatus, thereby affording a diastereomeric mixture of pyrrolidine-3,4-diesters **23a,b**. A portion of the mixture was subjected to column chromatography to separate the isomers for structure confirmation. After exhaustive reduction of the ester groups with LiAlH_4_, the mixture was oxidized with a H_2_O_2_/Na_2_WO_4_ system in aqueous methanol to give single nitrone **24** (Scheme 5). Spectral parameters of **24** are close to those of its 2,5,5-triethyl analog [20].

Addition of the nitrone to a solution of EtMgBr in THF leads to precipitation, presumably due to magnesium bis-alkoxide formation. Subsequent stirring of the reaction mixture with an excess of EtMgBr at room temperature or upon heating to reflux did not lead to Grignard reagent addition, and nitrone **24** was almost quantitatively recovered after the processing of the reaction mixture. To prevent precipitation of magnesium salts, the hydroxy groups were protected via treatment with trimethylsilyl chloride and triethylamine in dry THF (Scheme 6). Silylated nitrone **25** also showed no reactivity toward EtMgBr but reacted with *n*-BuLi. The reaction was complete after stirring at ambient temperature for 72 h, affording a mixture of nitroxides (presumably, diastereomers) with major isomer content exceeding 90%. The major isomer (racemate) was isolated via crystallization with the yield 65%, and the structure was determined by single-crystal X-ray analysis (Figure 1).

Along with nitroxide **5**, colorless oily compound **26** was isolated from the reaction mixture using chromatography. Analysis of ^1^H and ^13^C NMR spectra with ^1^H-^1^H and ^1^H-^13^C correlations (see Appendix A) allowed us to assign the 5-(*tert*-butyl)-2,2-diethyl-3-(hydroxymethyl)-4-pentyl-3,4-dihydro-*2H*-pyrrole structure to this compound.

### 2.3. Properties of the New Nitroxides

EPR spectra of the synthesized nitroxides are presented in Figure 2. Parameters of the EPR spectra of new nitroxides **4** and **5** are given in Table 1, and spectral parameters of **1** and **2** recorded under the same conditions are provided for comparison. The spectra of both new nitroxides were characterized by a triplet of doublets with hyperfine coupling (*hfc*) on nitrogen and hydrogen nuclei. This behavior is typical for 3-substituted or 3,4-disubstituted pyrrolidine nitroxides with geminal ethyl groups at position 2 or 5 [18,20]. The origin of this additional doublet splitting has been previously investigated for imidazolidine nitroxides, which have similar geometry [35]. It was shown that a repulsion with a substituent at a neighboring sp^3^-hybridized asymmetric center makes geminal ethyls favor a conformation with high spin density at one of four methylene hydrogens. Replacement of the ethyl with any other group or removal of the neighboring substituent may change conformational behavior with disappearance of the large *hfc*.

Unlike **18**, nitroxides **4** and **5** showed remarkable thermal stability. Heating of the samples of **4** and **5** at 100 °C for 80 min didn’t lead to significant decay of the nitroxides EPR signal (See Appendix A). Nitroxides **4** and **5** are rather lipophilic (their octanol–water partition coefficients are 1000 and 850, respectively). Their solubility allowed to register the EPR spectra, but the signal-to-noise ratio was not sufficient to follow the kinetics of the reduction. Moreover, dissolving them in an ultrasonic bath yielded unreproducible results, presumably due to the formation of unstable micellar solutions. For this reason, redox properties of **4** and **5** were investigated in 50% methanol, where stable 0.1 mM solutions can be prepared. The kinetics of nitroxide decay in the presence of 0.5 M ascorbate at pH 7.4 are shown in Figure 3. Second-order reduction rate constants of the nitroxides are listed in Table 1. The rate constants were obtained from the initial part of the kinetics when accumulation of dehydroascorbate anion was negligible [36]. Reduction of **1** and **2** under the same conditions was evaluated for comparison. Reduction of **1** and **2** proceeded much slower in 50% methanol than in aqueous PBS solution, showing a decrease in the rate constant by the factor of 20.8 and 3.2, respectively. It is known that a solvent can significantly affect the rate constant of chemical reactions [37]. The basic mechanism is the influence of solvent polarity on the stabilization energy of initial compounds as well as the energy level of a transition state. An increase in solvent polarity accelerates the rates of reactions where a charge develops in the activated complex, starting from a neutral or slightly charged reactant. The solvent effect is well known in the chemistry of reversible homolyses of alkoxyamines, where one of the reaction partners is a nitroxide radical [38].

The kinetics of reduction in 50% methanol were very similar among **1**, **2**, and **4**, whereas in water, the reduction of **1** proceeded significantly faster than that of **2** (see Appendix A). Presumably, this difference can be attributed to a difference in solvation of the nitroxide, ascorbate anion, and the transition state. While the observed difference for **2** follows the pattern of an ion–dipole interaction, the reaction of **1**, which is in the anionic form at pH 7.4, with ascorbate anion is complicated by Coulomb’s repulsion, which is more important in a solvent with lower polarity. The presented data indicate that nitroxide **5** is the most resistant to reduction.

## 3. Materials and Methods

### 3.1. General Information

2,2,5,5-Tetraethyl-3,4-bis(hydroxymethyl)pyrrolidin-1-oxyl (**2**) was prepared according to a published protocol [20]. 3-Carboxy-2,2,5,5-tetraethylpyrrolidin-1-oxyl (**1**) [18] was prepared as described in the patent [39], and the IR spectrum of the nitroxide is given in Appendix A (cf. [18]). A ^1^H NMR spectrum of the corresponding hydroxylamine trifluoroacetate is depicted in Appendix A. Commercially available reagents, such as *n*-butyllithium and lithium diisopropylamide solutions, dimethyl fumarate, 2-amino-3,3-dimethylbutanoic acid and 2-pentanone were used as received from Acros Organics B.V.B.A, Geel, Belgium. Solvents were dried by standard procedures (described in the literature) prior to use. The IR spectra were recorded on a Bruker Vector 22 FT-IR spectrometer (Bruker, Billerica, MA, USA) in KBr pellets (1:150 ratio) or in neat samples (for oily compounds). UV spectra were acquired on a HP Agilent 8453 spectrometer (Agilent Technologies, Santa Clara, CA, USA) in ethanol solutions (concentration ~10^−4^ M). ^1^H NMR spectra were recorded on a Bruker AV 300 (300.132 MHz), AM 400 (400.134 MHz), and AV 600 (600.300 MHz) spectrometers. ^13^C NMR spectra were recorded on a Bruker AV 300 (75.467 MHz), AM 400 (100.614 MHz), and AV 600 (151 MHz) spectrometers. All the NMR spectra were acquired for 5–10% solutions in CDCl_3_ or a CDCl_3_–CD_3_OD 1:4 mixture at 300 K using the signal of the solvent as a standard. To confirm the structure of stable nitroxides, ^1^H NMR spectra were recorded of the solutions of corresponding hydroxylamines prepared via reduction of the nitroxide samples (10–20 mg) with Zn powder in a CD_3_OD–CF_3_COOH 10:1 mixture, see subsection 3.2.12.

The structures of compounds **4**, **5**, and **12** were determined by single-crystal X-ray analysis (Appendix A). X-ray diffraction data were obtained on a Bruker P4 for **12** and on a Bruker Kappa Apex II CCD diffractometer for **4** and **5** with Mo Kα radiation (λ = 0.71073 Å) and a graphite monochromator. Absorption corrections were applied empirically using *SADABS* programs [40] for **4** and **5** and by the integration method for **12**. The structures were solved by direct methods and refined by the full-matrix least-squares method against all *F*^2^ in anisotropic approximation (besides the H atoms) using the *SHELXL2014*/7 software suite (Dept. of Structural Chemistry, University of Göttingen, Göttingen, Germany [41]). The H atoms’ positions were processed via the riding model, except for the positions of hydroxyl groups in **5** (refined independently). There are two independent molecules in the asymmetric unit of the cell for **4**. Datasets CCDC 1972133–1972135 contain the supplementary crystallographic data for this paper. These data can be obtained free of charge via http://www.ccdc.cam.ac.uk/cgi-bin/catreq.cgi or from the Cambridge Crystallographic Data Centre, 12 Union Road, Cambridge CB2 1EZ, UK; fax: (+44) 1223 336 033; or e-mail: deposit@ccdc.cam.ac.uk.

HPLC analyses were performed with an Agilent 1100 liquid chromatography system (Agilent Technologies, Santa Clara, CA, USA) equipped with a quaternary pump, online degasser, autosampler, and diode array detector. Chromatographic separations were carried out on a ZORBAX SB-C18 column (150 mm × 4.6 mm, 5.0 μm). The flow rate was 0.6 mL/min. Detection was performed at 238 nm. Acetonitrile, methanol and water were used to prepare the mobile phase.

The reactions were monitored by thin layer chromatography (TLC) on Sorbfil UV-254 (Imid ltd, Krasnodar, Russia) and DC-Alufolien (Merck KGaA, Darmstadt, Germany) plates with chloroform, chloroform–hexane, diethyl ether–hexane, and ethyl acetate–hexane mixtures as eluents. Kieselgel 60 (Macherey-Nagel GmbH & Co. KG, Düren, Germany), and alumina were utilized as sorbents for the column chromatography.

The EPR spectra in water were recorded on a Bruker ER-200D-SRC spectrometer in 100 µL quartz capillary for 0.1 mM solutions degassed via bubbling with argon. Spectrometer settings: frequency, 9.87 GHz; microwave power, 5.0 mW; modulation amplitude, 0.05 mT; time constant, 50 ms; and conversion time, 5.12 ms. The EPR spectra in water–methanol solutions and in toluene were recorded with an Elexsys E540 X-band spectrometer (Bruker, Billerica, MA, USA) in a 100 µL quartz capillary for 0.1–0.3 mM solutions, with the following spectrometer settings: field center, 351.600 mT; sweep range, 10 mT; modulation amplitude, 0.15 mT; microwave power, 2 mW; time constant, 10.24 ms; and scan time, 41 ms. The EasySpin software (Version 5.2.28, easyspin.org, [42]) was employed for simulation of spectra. For kinetic measurements, the EPR spectra were acquired at the same instrument settings but with a greater modulation amplitude (0.2 mT) to optimize the signal-to-noise ratio. Thermal stability of nitroxides **4** and **5** was studied in 1 mM solutions in toluene within a sealed 100 µL glass capillary using a double sample resonator (Bruker, ER 4105DR). One of the two samples with the nitroxide was placed into a water bath (95–100 °C) for 80 min incubation, and nitroxide decay was followed via comparison of intensities. Partition coefficients in a water–octanol mixture were estimated from the amplitudes of the EPR spectra of a nitroxide in water after extensive shaking with different portions of added octanol and separation.

For kinetic measurements in water, stock solutions of the nitroxide, ascorbic acid, and of glutathione in phosphate buffer (1 mM, pH 7.4) were prepared, and pH was adjusted to 7.4 with NaHCO_3_. All the solutions were deoxygenated with argon, carefully and quickly mixed in a small tube to attain final concentrations (nitroxide, 0.1–0.3 mM; GSH, 2 mM; and ascorbate, 166.7 mM) and were placed into an EPR capillary (50 μL). The capillary was sealed on both sides and placed into the EPR resonator. Alternatively, all the reagents were dissolved in a MeOH–H_2_O mixture (1:1), pH of the stock solutions was adjusted to 7.4 with NaHCO_3_, and the solutions were mixed in a similar manner to obtain final concentrations: nitroxide, 0.1 mM; GSH, 2 mM; and ascorbate, 500 mM. The decay of amplitude of the low-field component of the EPR spectrum was followed to obtain the kinetics. Initial part of the decay curves (up to 200 min) was used for fitting. Kinetics of the decay was fitted to a monoexponential function to calculate the first-order rate constants. Then, these constants were divided by the concentration of ascorbic acid to calculate the second-order reaction constants.

### 3.2. Synthesis

#### 3.2.1. 5,5-Diethyl-1-hydroxy-2,2,4-trimethyl-2,5-dihydro-1*H*-imidazole (**7**)

It was prepared from 3-hydroxyamino-3-ethylpentan-2-one hydrochloride [16] and acetone as described previously [27].

#### 3.2.2. (*Z*)-4-(3,3-Dimethyl-2-oxobutylidene)-5,5-diethyl-2,2-dimethylimidazolidin-1-oxyl (10)

A solution of 4.3 g (23.4 mmol) of imidazoline **6** in 30 mL of dry diethyl ether was added dropwise to 35 mL of a 2 M solution of lithium diisopropylamide in an argon atmosphere. After 50 min of stirring, ethyl pivalate (12 mL, 78.5 mmol) was added, and the reaction mixture was heated to reflux for 48 h. Then, the mixture was cooled to 5 °C, and 100 mL of water was added carefully. The organic layer was separated, and the aqueous layer was extracted with diethyl ether (30 mL × 3 times). The combined extract was agitated with 10 g of manganese dioxide for 1 h and dried over magnesium sulfate. The precipitate was filtered off, and the filtrate was evaporated under reduced pressure. The crude product was purified by column chromatography on silica gel with gradient elution (from hexane to a diethyl ether–hexane 1:4 mixture) to give **10** (5.5 g, 88% yield) as a yellow crystalline solid, m.p. 84–86 °C (hexane). Elemental analysis: found: C, 67.66; H, 10.18; N, 10.46; calcd. for C_15_H_27_N_2_O_2_: C, 67.38; H, 10.18; N, 10.48%; IR (KBr) ν_max_: 3274, 2978, 2952, 2869, 1636, 1561, 1494, 1461, 1444, 1377, 1362, 1328, 1217, 1181, 1163, 1121, 1018, 968, 947, 874, 815, 793, 748, cm^−1^; UV (EtOH) λ_max_ (log ε): 298 (4.27).

#### 3.2.3. (*Z*)-1-(5,5-Diethyl-2,2-dimethyl-1-hydroxyimidazolidin-4-ylidene)-3,3-dimethylbutan-2-one (**9**)

Hydroxyamine **9** was prepared via hydrogenation of **10** on a Pd/C catalyst in methanol under atmospheric pressure and ambient temperature. Yield 97%, colorless crystalline solid, m.p. 108–111 °C (hexane). Elemental analysis: found: C, 68.03; H, 10.58; N, 9.79; calcd. for C_15_H_28_N_2_O_2_: C, 67.13; H, 10.52; N, 10.44%; IR (KBr) ν_max_: 3281 (br.), 2968, 2871, 1617, 1526, 1461, 1392, 1378, 1361, 1341, 1327, 1216, 1198, 1170, 1126, 1043, 1020, 988, 942, 910, 875, 812, 784, 752, 701, cm^−1^; UV (EtOH) λ_max_ (log ε): 304 (4.28); ^1^H NMR(300 MHz; CDCl_3_, δ): 0.92 (t, *J* = 7.3 Hz, 6 H), 1.15 (s, 9H), 1.42 (s, 6H,), 1.69, 1.82 (ABq, *J*_q_ = 7.3 Hz, *J*_AB_ = 14.5 Hz, both 2H), 5.01 (s, 1H), 5.35 (br. s, 1H), 10.12 (br. s, 1H); ^13^C{^1^H} NMR (75 MHz; CDCl_3_, δ): 8.94, 27.57, 27.80, 27.83, 41.41, 74.03, 79.97, 83.91, 165.45, 204.83.

#### 3.2.4. Enaminoketone 9 Acid Hydrolysis and Recyclization

Method A: Similarly to a previously described procedure, [25] a solution of 2.33 g (8.7 mmol) of enaminoketone **9** in methanol (10 mL) was diluted with 35% hydrochloric acid (10 mL), and the reaction mixture was kept for 84 h at ambient temperature. Then, methanol was removed under reduced pressure, and the mixture was neutralized with a saturated aqueous solution of sodium carbonate under argon and extracted with degassed chloroform (10 mL × 3 times). The combined extract was dried over magnesium sulfate and evaporated under reduced pressure. The residue was purified by recrystallization from hexane to isolate **6** in a 1.1 g (60%) yield.

Method B: Concentrated hydrochloric acid (40 mL) was added to a solution of 7.6 g (28.7 mmol) of **9** in 40 mL of methanol, and the reaction mixture was kept for 40 h at room temperature. Neutralization of the reaction mixture resulted in precipitation of colorless oil, which was separated and dissolved in chloroform, washed with brine and sodium carbonate, and dried over magnesium sulfate. Next, chloroform was removed under reduced pressure, and the residue was triturated with hexane. The formed precipitate was filtered off and recrystallized from hexane to obtain **13** (0.9 g, 15%). The aqueous solution after neutralization was processed as described in method A to obtain **6** (2.9 g, 47%). Compound **13** totally decomposed in a few days under aerobic conditions at room temperature. The mixture of decomposition products was separated by column chromatography on silica gel using a chloroform–hexane 1:4 mixture as an eluent to isolate **16** and **17**.

*5-(tert-Butyl)-2,2-diethyl-3-oxo-3,4-dihydro-2H-pyrrole 1-oxide (**6**):* A colorless crystalline solid, m.p. 138–143 °C (hexane). Elemental analysis: found: C, 67.83; H, 10.36; N, 6.32; calcd. for C_12_H_21_NO_2_: C, 68.21; H, 10.02; N, 6.63%; IR (KBr) ν_max_: 2964, 2924, 2877, 2580 (br.), 1594, 1534 (br.), 1495, 1427, 1392, 1361, 1323, 1273, 1253, 1217, 1135, 1122, 1066, 1028, 951, 868, 792, 764, cm^−1^; UV (EtOH) λ_max_ (log ε): 332 (3.97); ^1^H NMR (300 MHz; CDCl_3_, δ): Form A: 0.67 (br. m, 6H), 1.35 (s, 9H), 1.67, 1.92 (br. ABq, *J_q_* = 6.8 Hz, *J_AB_* = 13.6 Hz, both 2H), 3.17 (br. s, 2H); Form B: 0.66 (br. m, 6H,), 1.35 (s, 9H), 1.67, 1.81 (br. ABq, *J_q_* = 6.8 Hz, *J_AB_* = 13.6 Hz, both 2H), 4.96 (br. s, 1H), 10.22 (br. s, 1H); ^13^C{^1^H} NMR (75 MHz; CDCl_3_, δ): Form A: 7.64, 25.64, 28.14, 33.89, 42.87, 85.06, 151.31, 209.84; Form B: 7.64, 27.92, 34.38, 79.50, 94.50, 185.96, 196.55.

*3-Amino-5-(tert-butyl)-2,2-diethyl-2H-pyrrole 1-oxide (**13**)*: A yellow crystalline solid. IR (KBr) ν_max_: 3320, 3150 (br.), 2972, 2877, 2737, 1661, 1588, 1484, 1459, 1407, 1378, 1359, 1337, 1299, 1255, 1179, 1149, 1090, 1055, 951, 870, 788, 763, 706, 674, 621, cm^−1^; ^1^H NMR (300 MHz; CDCl_3_, δ): 0.55 (t, *J* = 7.2 Hz, 6H), 1.26 (s, 9H), 1.40, 1.92 (ABq, *J_q_* = 7.2 Hz, *J_AB_* = 13.8 Hz, both 2H), 4.63 (br. s, 2H), 5.06 (s, 1H); ^13^C{^1^H} NMR (75 MHz; CDCl_3_, δ): 7.21, 26.54, 28.70, 33.90, 77.01, 91.70, 153.59, 159.09.

*5-(tert-Butyl)-2,2-diethyl-3-imino-4-oxo-3,4-dihydro-2H-pyrrole 1-oxide (**16**):* Yellow oil. IR (neat) ν_max_: 3369, 3207, 2972, 2935, 2882, 2774, 1771, 1703, 1678, 1624, 1588, 1570, 1510, 1481, 1456, 1410, 1382, 1359, 1322, 1271, 1230, 1202, 1147, 1121, 1068, 1040, 1006, 964, 947, 864, 821, 791, 770, 753, 705, cm^−1^; M/z, found: M 224, (M-CH_3_) 209; calcd. for C_12_H_20_N_2_O_2_: 224; ^1^H NMR (400 MHz; CDCl_3_, δ): 0.63 (t, *J* = 7.4 Hz, 6H), 1.39 (s, 9H), 1.93, 2.08 (ABq, *J_q_* = 7.4 Hz, *J_AB_* = 14.4 Hz, both 2H); ^13^C{^1^H} NMR (100 MHz; CDCl_3_, δ): 7.02, 25.71, 28.71, 33.94, 81.65, 153.93, 171.42, 178.29.

*5-(tert-Butyl)-2,2-diethyl-3,4-dioxo-3,4-dihydro-2H-pyrrole 1-oxide (**17**)*: yellow crystalline solid, m.p. 31–31 °C (hexane). Elemental analysis: found: C, 64.29; H, 8.49; N, 6.49; calcd. for C_12_H_19_NO_3_: C, 63.98; H, 8.50; N, 6.22%; m/z, Found: M 225, (M-CH_3_) 210; calcd. for C_12_H_19_NO_3_: M 225; IR (KBr) ν_max_: 2927, 2855, 2769, 1769, 1696 (br.), 1503, 1454, 1408, 1381, 1361, 1320, 1226, 1192, 1120, 1085, 1039, 931, 870, 805, 772, 722, 687, cm^−1^; UV (EtOH) λ_max_ (log ε): 239 (3.83), 328 (3.96); ^1^H NMR (400 MHz; CDCl_3_, δ): 0.69 (t, *J* = 7.4 Hz, 6H), 1.43 (s, 9H), 1.88, 2.00 (ABq, *J_q_* = 7.4 Hz, *J_AB_* = 14.2 Hz, both 2H); ^13^C{^1^H} NMR (100 MHz; CDCl_3_, δ): 7.28, 25.71, 27.25, 34.41, 81.70, 159.63, 178.56, 197.35.

#### 3.2.5. 3,3’-Bis(2-*tert*-butyl-5,5-diethyl-4-oxopyrrolinylidene) 1,1’-dioxide (**12**)

Manganese dioxide 0.75 g (8.63 mmol) was added to a solution of 0.15 g (0.71 mmol) of **6** in chloroform (3 mL), and the reaction mixture was stirred vigorously for 7 days. The precipitate was filtered off, and the filtrate was evaporated under reduced pressure. The residue was recrystallized from ethyl acetate to give 0.145 g (95%) of dimer **12** as a dark violet crystalline solid, m.p. 157–160 °C (ethyl acetate). Elemental analysis: found: C, 68.91; H, 9.27; N, 6.71; calcd. for C_24_H_38_N_2_O_4_: C, 68.87; H, 9.15; N, 6.69%; IR (KBr) ν_max_: 2972, 2928, 2850, 1701, 1505, 1489, 1459, 1422, 1379, 1359, 1298, 1219, 1118, 1050, 953, 879, 859, 778, 754, 717, 668, cm^−1^; UV (EtOH) λ_max_ (log ε): 234 (3.83), 307 (3.93), 375 (3.70), 579 (4.03); ^1^H NMR (400 MHz; CDCl_3_, δ): 0.44, 0.63 (t, *J* = 7.35 Hz, both 6H), 1.39 (s, 18H), 1.66, 1.75 (ABq, *J_q_* = 7.3 Hz, *J_AB_* = 14.0 Hz both 2H), 1.74, 1.89 (ABq, *J_q_* = 7.4 Hz, *J_AB_* = 14.6 Hz, both 2H). ^13^C{^1^H} NMR (100 MHz; CDCl_3_, δ): 7.07, 7.39, 26.01, 27.72, 29.39, 36.03, 82.02, 198.35.

#### 3.2.6. 5-(*tert*-Butyl)-5-butyl-2,2-diethyl-3-oxopyrrolidin-1-oxyl (**18**)

A warm solution of 0.5 g (2.37 mmol) of pyrroline **6** in dry benzene (30 mL) was added dropwise to a 2.6 M toluene solution of BuLi (14 mL) in an argon atmosphere. The reaction mixture was heated at 60 °C for 1 h and then kept at room temperature for 24 h. After that, the mixture was cooled to 0–5 °C and quenched with 30 mL of water. The organic layer was separated, and the aqueous layer was extracted with benzene. The combined extract was dried over magnesium sulfate and evaporated under reduced pressure. The residue was purified by column chromatography on silica gel with gradient elution (from tetrachloromethane to a chloroform–tetrachloromethane 1:5 mixture) to isolate **18** (0.39 g, 61%) as a yellow crystalline solid, m.p. 42–43 °C (hexane). Elemental analysis: found: C, 71.90; H, 11.49; N, 4.96; calcd. for C_16_H_30_NO_2_: C, 71.59; H, 11.27; N, 5.22%; HRMS (EI/DFS) m/z [M]^+^ calcd for (C_16_H_30_NO_2_)^+^: 268.2271; found: 268.2276. IR (KBr) ν_max_: 2969, 2874, 1747, 1483, 1470, 1407, 1396, 1379, 1366, 1337, 1297, 1277, 1213, 1161, 1148, 1109, 927, 888, 839, 785, 733, cm^−1^. UV (EtOH) λ_max_ (log ε): 217 (3.22), 243 (3.19). HPLC UV 99.0355%.

#### 3.2.7. Thermal Decomposition of Nitroxide 18 in the Presence of TEMPO

A mixture of 0.3 g (1.12 mmol) of nitroxide **18** and 1.6 g (10.3 mmol) of TEMPO was placed into a Schlenk flask and heated at 70–80 °C in vacuum for 7 days. The mixture was separated by column chromatography on silica gel with gradient elution (from hexane to a diethyl ether–hexane 1:2 mixture) to give **20** (0.09 g, 38%) and **21** (0.2 g, 85%).

*1-(tert-Butoxy)-2,2,6,6-tetramethylpiperidine (**20**)*: Colorless oil. NMR ^1^H and ^13^C spectra match the literature data [43,44].

*3,3’-Bis(2-butyl-5,5-diethyl-4-oxopyrrolinylidene) 1,1’-dioxide (**21**)*: A dark crimson crystalline solid, m.p. 95–100 °C (hexane). Elemental analysis: found: C, 69.06; H, 8.98; N, 6.55; calcd. for C_24_H_38_N_2_O_4_: C, 68.87; H, 9.15; N, 6.69%; IR (KBr) ν_max_: 2954, 2929, 2872, 1708, 1461, 1409, 1345, 1315, 1232, 1118, 1051, 935, 712, cm^−1^. UV (EtOH) λ_max_ (log ε): 228 (3.89), 284 (3.71), 341 (3.81), 508 (4.19); ^1^H NMR (300 MHz; CDCl_3_, δ): 0.65 (t, *J* = 7.4 Hz, 3H), 0.86 (t, *J* = 7.4 Hz, 3H), 0.91 (t, *J* = 7.4 Hz, 3H), 1.39 (sex, *J* = 7.4 Hz, 2H), 1.60 (m, 2H,), 1.83, 1.94 (ABq, *J_AB_* = 14.3 Hz, *J_q_* = 7.4 Hz, 4H), 2.95 (m, 2H); ^13^C{^1^H} NMR (75 MHz; CDCl_3_, δ): 7.02, 13.65, 22.24, 25.97, 27.14, 27.65, 80.57, 122.45, 159.55, 199.05.

#### 3.2.8. 5-(*tert*-Butyl)-5-butyl-2,2-diethyl-3-hydroxypyrrolidin-1-oxyl (**4**)

A 1.3-fold excess of sodium borohydride (0.037 g, 0.97 mmol) was added in portions to a solution of nitroxide **18** (0.2 g, 0.75 mmol) in ethanol with stirring for 30 min. After that, ethanol was removed under reduced pressure, and the residue was diluted with an equal volume of water and extracted with diethyl ether. The extract was dried over magnesium sulfate and then evaporated under reduced pressure. The residue was recrystallized from hexane. Nitroxide **4** was isolated in a 0.197 g (98%) yield as a pale yellow crystalline solid, m.p. 88–90 °C (hexane). Elemental analysis: found: C 70.76; H 11.80; N 5.09%; calcd. for C_16_H_32_NO_2_, C 71.06; H 11.93, N 5.18%. HRMS (EI/DFS) m/z [M]^+^ calcd for (C_16_H_32_NO_2_)^+^: 270.2428; found: 270.2426. IR (KBr) ν_max_: 3392 (br.), 2963, 2936, 2877, 1461, 1405, 1383, 1367, 1311, 1282, 1221, 1144, 1122, 1095, 1056, 933, 829, 746, 671, cm^−1^. UV (EtOH) λ_max_ (log ε): 236 (3.32). HPLC UV 96.2762%.

#### 3.2.9. 5-(tert-Butyl)-3,4-bis(methoxycarbonyl)-2,2-diethylpyrrolidine (**23a,b**)

A mixture of 2-amino-3,3-dimethylbutanoic acid (1.33 g, 10 mmol), dimethyl fumarate (1.5 g, 10 mmol), diethylketone (10 mL, 100 mmol), DMF (10 mL), and toluene (10 mL) was placed into a Dean–Stark apparatus and stirred under reflux for 3 days. The solvent was distilled off in vacuum, and the residue was dissolved in ethyl acetate (30 mL). The solution was washed with aqueous sodium bicarbonate (50 mL × 3 times) and extracted with 5% sulfuric acid (20 mL × 3 times). The acidic extracts were basified with Na_2_CO_3_ to pH 7–8 and extracted with ethyl acetate (20 mL × 3 times). The extract was dried with Na_2_CO_3_, and the solvent was removed under reduced pressure to give 2.2 g (73%) of a crude diastereomeric mixture as yellow oil. The mixture was used in the next step without further purification. To confirm the structure, the isomers were separated by column chromatography on silica gel (eluent: hexane–ethyl acetate 50:1).

*Isomer **23a***: Yield 0.6 g (20%), colorless crystalline solid, m.p. 30.3–32.0 °C (hexane). Elemental analysis: found: C, 64.00; H, 9.76; N, 4.60; calcd. for C_16_H_29_NO_4_: C, 64.18; H, 9.76; N, 4.68%; HRMS (EI/ Agilent 7200 Accurate Mass Q-TOF GC/MS) m/z [M-CH_3_O]^+^ calcd. for C_15_H_26_NO_3_ 268.1907, found 268.1913.IR (KBr) ν_max_: 3444, 2953, 2879, 1724, 1620, 1460, 1435, 1385, 1369, 1323, 1277, 1259, 1194, 1169, 1128, 1065, 1039, 1016, 974, 935, 893, 858, 804, 779, cm^−1^. ^1^H NMR (500 MHz; CDCl_3_, δ): 0.77 (t, *J* = 7.3 Hz, 3H), 0.88 (s, 9H), 0.93 (t, *J* = 7.5 Hz, 3H), 1.12 (dq, *J_d_* = 14.0 Hz, *J_q_* = 7.3 Hz, 1H), 1.38 (dq, *J_d_* = 14.0 Hz, *J_q_* = 7.3 Hz, 1H), 1.64 (br, 1H), 1.66 (dq, *J_d_* = 13.6 Hz, *J_q_* = 7.5 Hz, 1H), 1.70 (dq, *J_d_* = 13.6 Hz, *J_q_* = 7.5 Hz, 1H), 2.98 (d, *J* = 6.3 Hz, 1H), 3.02 (d, *J* = 3.0 Hz, 1H), 3.29 (dd, *J_1_* = 6.3 Hz, *J_2_* = 3.0 Hz, 1H), 3.57 (s, 3H), 3.58 (s, 3H). ^13^C{1H} NMR (125 MHz, CDCl_3_, δ): 8.3, 8.4, 24.8, 27.2, 29.8, 32.4, 50.1, 51.29, 51.33, 57.3, 67.9, 70.8, 173.4, 175.7.

*Isomer **23b***: Yield 1.36 g (46%), colorless oil. Elemental analysis: found: C, 64.17; H, 9.73; N, 4.68; calcd. for C_16_H_29_NO_4_: C, 64.18; H, 9.76; N, 4.68%. HRMS (EI/ Agilent 7200 Accurate Mass Q-TOF GC/MS) m/z [M-CH_3_O]^+^ calcd. for C_15_H_26_NO_3_ 268.1907, found 268.1906. IR (neat) ν_max_: 2955, 2879, 1738, 1462, 1437, 1375, 1338, 1254, 1194, 1167, 1107, 1066, 1020, 982, 966, 820, 785, 744, cm^−1^. ^1^H NMR (500 MHz; CDCl_3_, δ): 0.79 (t, *J* = 7.4 Hz, 3H), 0.85 (s, 9H), 0.87 (t, *J* = 7.3 Hz, 3H), 1.25 (dq, *J_d_* = 13.9 Hz, *J_q_* = 7.4 Hz, 1H), 1.31 (dq, *J_d_* = 13.9 Hz, *J_q_* = 7.4, 1H), 1.36 (br, 1H), 1.56 (dq, *J_d_* = 14.3 Hz, *J_q_* = 7.3 Hz, 1H), 1.60 (dq, *J_d_* = 14.3 Hz, *J_q_* = 7.3 Hz, 1H), 3.04 (d, *J* = 8.8 Hz, 1H), 3.07 (dd, *J_1_* = 8.8 Hz, *J_2_* = 8.4 Hz, 1H), 3.19 (d, J = 8.4 Hz, 1H), 3.61 (s, 3H), 3.61 (s, 3H). ^13^C{1H} NMR (125 MHz, CDCl_3_, δ): 7.8, 7.9, 26.3, 28.6, 29.3, 33.2, 48.9, 51.3, 51.6, 57.3, 66.3, 70.1, 173.0, 175.3.

#### 3.2.10. 5-(*tert*-Butyl)-2,2-diethyl-3,4-bis(hydroxymethyl)-3,4-dihydro-2*H*-pyrrole 1-oxide (**24**)

A solution of a crude mixture of amines **23a,b** (6.38 g, 22 mmol) in dry diethyl ether (20 mL) was added dropwise to a stirred solution of LiAlH_4_ (1.56 g, 41 mmol) in dry diethyl ether (100 mL). The mixture was stirred at reflux for 1 h, then the flask was cooled in an ice bath and carefully quenched with 5 mL of 5% aqueous sodium hydroxide and 15 mL of water. The organic layer was separated via decantation, the wet precipitate was washed with diethyl ether (20 mL × 3 times), and the combined extract was evaporated under reduced pressure. The residue was dissolved in methanol (50 mL), mixed with a solution of sodium tungstate (0.7 g, 2.12 mmol) and EDTA disodium salt (0.71 g, 2.12 mmol) in water (30 mL), and hydrogen peroxide 30% (7 mL) was added. The solution was kept at ambient temperature for 3 days, then a catalytic amount of manganese dioxide (0.1 g, 1.2 mmol) was carefully added for quenching of remaining H_2_O_2_. After oxygen evolution ceased, the solution was evaporated in vacuum. The residue was triturated with chloroform, and the extract was dried with sodium carbonate. The solution was filtered, and the solvent was distilled off in vacuum. The residue was triturated with diethyl ether to give 4.16 g of crude **24** as a pinkish crystalline solid. The stock solution was evaporated under reduced pressure and purified by column chromatography on silica gel (eluent: methanol–ethyl acetate 1:100) to afford another portion of **24 (**0.49 g).

***24***: Yield 4.65 g (85%), colorless crystals, m.p. 121.6–122.4 °C (ethyl acetate). Elemental analysis: found: C, 65.61; H, 10.83; N, 5.46; calcd. for C_14_H_27_NO_3_: C, 65.33; H, 10.57; N, 5.44%. HRMS (EI/DFS) m/z [M]^+^ calcd for (C_14_H_27_NO_3_)^+^ 257.1986, found 257.1985. IR (KBr) ν_max_: 3404, 3194, 2972, 2949, 2941, 2922, 2881, 2739, 1711, 1569,1487, 1454, 1396, 1373, 1331, 1292, 1242, 1157,1126, 1095, 1086, 1057, 1043, 1030, 1005, 943, 928, 901, 862, 796, 779, 739, 692, 669, 602, 575, 555, 486, 471, 444, 436, cm^−1^. UV (EtOH) λ_max_ (log ε): 239 (3.97). ^1^H NMR (400 MHz; CDCl_3_, δ): 0.70 (t, *J* = 7.3 Hz, 3H), 0.83 (t, *J* = 7.4 Hz, 3H), 1.29 (s, 9H), 1.48 (dq, *J_d_* = 14.5 Hz, *J_q_* = 7.3 Hz, 1H), 0.59 (dq, *J_d_* = 14.5 Hz, *J_q_* = 7.3 Hz, 1H), 1.67 (dq, *J_d_* = 14.5 Hz, *J_q_* = 7.4 Hz, 1H), 1.75 (dq, *J_d_* = 14.5 Hz, *J_q_* = 7.3 Hz, 1H), 2.29 (ddd, *J_1_* = 5.9 Hz, *J_2_* = 10.0 Hz, *J_3_* = 4.7 Hz, 1H), 2.83 (ddd, *J_1_* = 5.9 Hz, *J_2_* = 8.9 Hz, *J_3_* = 3.0, 1H), 3.33 (t, *J* = 9.1 Hz, 1H), 3.61 (t, *J* = 9.3 Hz, 1H), 3.69 (br, 1H), 4.10 (br, 1H), 5.14 (br, 1H), 5.24 (br, 1H). ^13^C{1H} NMR (100 MHz, CDCl_3_, δ): 7.4, 9.1, 26.2, 27.3, 31.4, 34.4, 44.5, 50.6, 61.3, 64.8, 80.8, 152.9.

#### 3.2.11. 2-(tert-Butyl)-2-butyl-5,5-diethyl-3,4-bis(hydroxymethyl)pyrrolidin-1-oxyl (5)

A solution of trimethylsilyl chloride (1.54 g, 14.1 mmol) in dry THF (10 mL) was added dropwise to a solution of nitrone **24** (1.65 g, 6.42 mmol) and triethylamine (1.3 g, 28.2 mmol) in dry THF (35 mL) upon stirring in ice bath. Then, the solvent was evaporated under reduced pressure, the residue was triturated with diethyl ether (10 mL × 5 times), and the precipitate was filtered off. The combined filtrate was concentrated under reduced pressure and dissolved in dry benzene (20 mL). Next, a solution of *n*-BuLi (1 M in hexane, 30 mL) was added in an argon atmosphere upon stirring in an ice bath. The reaction mixture was kept at ambient temperature for 3 days. The reaction was controlled with TLC on SiO_2_ (eluent: ethyl acetate–hexane 1:4), using H_3_[P(Mo_3_O_10_)_4_]⋅nH_2_O for staining. The mixture was quenched carefully with water until formation of two phases. The organic layer was separated, and the aqueous layer was extracted with ethyl acetate (15 mL × 2 times). The combined extract was evaporated under reduced pressure, the residue was dissolved in methanol (20 mL), and an aqueous solution of PPTS (0.05 g in 3 mL) was added to remove the protective groups. The mixture was kept in ambient air for 3 days, and then methanol was removed under reduced pressure, and the residue was diluted with water (10 mL) and extracted with ethyl acetate (10 mL × 4 times). The extract was evaporated under reduced pressure, and the residue was separated by column chromatography on silica gel (eluent: ethyl acetate–hexane 1:4) to give **5** (1.31 g, 65%) and **26** (0.36 g, 20%). For purification, nitroxide **5** was recrystallized from a hexane–ethyl acetate (6:1) solution and then from an ethanol–water (2:1) solution.

*2-(tert-Butyl)-2-butyl-5,5-diethyl-3,4-bis(hydroxymethyl)pyrrolidin-1-oxyl (**5**)*: A yellow crystalline solid, m.p. 109.4–113.8 °C (hexane–ethyl acetate 6:1). Elemental analysis: found: C, 68.90; H, 11.55; N, 4.58; calcd. for C_18_H_36_NO_3_: C, 68.75; H, 11.54; N, 4.45%. HRMS (EI/DFS) m/z [M]^+^ calcd. for (C_18_H_36_NO_3_)^+^ 314.2690, found 314.2688. IR (KBr) *ν*_max_: 3333, 2962, 2875, 1485, 1460, 1425, 1392, 1379, 1367, 1346, 1298, 1288, 1201, 1163, 1138, 1063, 1051, 1012, 995, 949, 922, 839, 795, 754, 739, 710, 604, 586, 478 cm^−1^. UV (EtOH), λ_max_ (log ε): 239 (3.23). HPLC UV 96.8533%.

*5-(tert-Butyl)-2,2-diethyl-3-(hydroxymethyl)-4-pentyl-3,4-dihydro-2H-pyrrole (**26**)*: colorless oil. Elemental analysis: found: C, 76.38; H, 12.08; N, 4.43; calcd. for C_18_H_35_NO: C, 76.81; H, 12.53; N, 4.98%. HRMS (EI/DFS) m/z [M]^+^ calcd for (C_18_H_35_NO)^+^ 281.2713, found 281.2710.IR (neat) *ν*_max_: 3377, 2960, 2933, 2874, 1626, 1462, 1392, 1363, 1219, 1201, 1167, 1092, 1059, 1018, 955, 924, 837, 802, 727, 582 cm^−1^. UV (EtOH) λ_max_ (log ε): 204 (3.01), 232 (2.52). ^1^H NMR (600 MHz; CDCl_3_, δ): 0.81 (t, *J_t_* = 7.4 Hz, 3H), 0.84 (t, *J_t_* = 7.4 Hz, 3H), 0.85 (t, *J_t_* = 7.0 Hz, 3H), 1.16 (s, 9H), 1.17–1.41 (m, 7H), 1.41 (dq, *J_d_* = 14.2 Hz, *J_q_* = 7.2 Hz, 1H), 1.44 (dq, *J_d_* = 14.0 Hz, *J_q_* = 7.4 Hz, 1H), 1.53 (dq, *J_d_* = 14.0 Hz, *J_q_* = 7.4 Hz, 1H), 1.61 (dq, *J_d_* = 14.2 Hz, *J_q_* = 7.2 Hz, 1H), 1.84 (m, 1H), 1.96 (ddd, *J_d1_* = 7.0 Hz, *J_d2_* = 7.0 Hz, *J_d3_* = 6.3 Hz, 1H), 2.64 (ddd, *J_d1_* = 5.9 Hz, *J_d2_* = 10.0 Hz, *J_d3_* = 4.7 Hz, 1H), 3.47 (dd, *J_d1_* = 10.6 Hz, *J_d2_* = 6.8 Hz, 1H), 3.67 (dd, *J_d1_* = 10.6 Hz, *J_d2_* = 6.8 Hz, 1H). ^13^C{^1^H} NMR (150 MHz, CDCl_3_, δ): 8.33, 8.96, 13.89, 22.44, 27.30, 27.36, 29.35, 31.86, 32.62, 33.68, 35.98, 50.21, 53.64, 63.32, 76.46, 181.82. For 2D NMR correlation spectra ^1^H−^1^H (COSY, mixing time 0.8 s) and ^1^H−^13^C (HSQC, HMBC) see Appendix A.

#### 3.2.12. The General Method for Nitroxides 4 and 5, Reduction with Zn in CF_3_COOH for NMR

Trifluoroacetic acid (20–50 μL) was added dropwise to a mixture of of the nitroxide (0.04–0.08 mmol), Zn dust (50–100 mg, 0.8–1.5 mmol) and CD_3_OD (0.3–0.4 mL). The reaction mass was incubated at ambient temperature for 15 min and then diluted with 0.3 mL of CDCl_3_; the precipitate was filtered off. The filtrate was placed into an NMR tube, and ^1^H spectra were recorded.

*5-(tert-Butyl)-5-butyl-2,2-diethylpyrrolidine-1,3-diol 2,2,2-trifluoroacetate (**4H**)*: ^1^H NMR (400 MHz; CDCl_3_, CD_3_OD, CF_3_COOH, δ): 0.94 (t, *J* = 6.8 Hz, 3H), 1.03, 1.05 (t, *J* = 7.4 Hz, both 3H), 1.08 (s, 9H), 1.23 (m, 1H), 1.36 (m, 4H), 1.66 (m, 2H), 1.78-2.06 (m, 4H), 2.12 (m, 1H), 2.23 (m, 1H), 4.31 (m, 1H).

*(2-(tert-Butyl)-2-butyl-5,5-diethyl-1-hydroxypyrrolidine-3,4-diyl)dimethanol 2,2,2-trifluoroacetate (**5H**)*: ^1^H NMR (400 MHz; CDCl_3_, CD_3_OD, CF_3_COOH, δ): 0.97 (t, *J* = 6.8 Hz, 3H), 1.03, 1.07 (t, *J* = 7.4 Hz, both 3H), 1.15 (s, 9H), 1.26–1.51 (m, 4H), 1.68–2.01 (m, 6H), 2.35 (m, 1H), 2.48 (m, 1H), 3.66 (m, 1H), 3.76 (m, 1H), 3.87 (m, 2H).

## 4. Conclusions

Here, we again demonstrated that the addition of organolithium compounds to cyclic α-*tert*-butyl nitrones is an efficient method of the synthesis of highly strained nitroxides. The new 2-butyl-2-*tert*-butyl pyrrolidine-1-oxyls were prepared in good yields. The data on reduction of the new nitroxides mean that a further increase in the steric demand of the substituents will make the nitroxides more resistant to reduction. Modification of the hydroxy groups may enable incorporation of these nitroxides into macromolecular or supramolecular nanostructures that can serve as organic radical contrast agents for in vivo MRI. Examples of similar application of highly lipophilic nitroxides were mentioned in the introduction. The use of **5** and similar structures with exceptionally high resistance to reduction can ensure long lifetime of paramagnetic centers in live tissues and a prolonged enhancement for MRI.

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
