# Peer review of "2-Butyl-2-tert-butyl-5,5-diethylpyrrolidine-1-oxyls: Synthesis and Properties"

_molecules, 2020, doi:10.3390/molecules25040845_

Round 1

Reviewer 1 Report

The authors describe in their manuscript the synthesis of two new nitroxides with increased steric shielding by introducing n-butyl and tert-butyl groups at  position 2. All compounds are fully characterized and sufficiently pure. The synthesis involves several steps and is elaborate. In addition, the authors test the thermal and reduction stability of both nitroxides. They find that compared to previous attempts with two tert-butyl groups in position 2, both new nitroxides are thermally stable up to at least 100 °C. They also find increased stability with respect to the already rather reduction stable gem-diethyl nitroxides, which is exciting. Thus, the manuscript is interesting to the readership of Molecules and I recommend its acceptance.

May I just suggest to add one reference in the introduction, where the authors talk about the reduction stability of gem-diethyl nitroxides. Exactly this has been very recently explored in a paper by Wuebben et al. for such nitroxides attached to DNA and published in Molecules 2019, 24, 4482.

Reviewer 2 Report

This manuscript describes the full details of the preparation, thermal stability, and kinetic reduction of new pyrrolidine-1-oxyls.

It is well accepted that N-oxyls are important not only for organic reactions as an oxidant, but also for the biomedical researches as a molecule probe.  The authors demonstrated the detailed synthetic schemes for the new pyrrolidine-1-oxyls having a variety of substituents at 2, 5 positions by using the nucleophilic addition of n-butyllithium instead of Grignard reagents.  The low reactivity against Grignard reagents could be a limitation of this synthetic methodology, but does not decrease the importance and usefulness of this research.

Chemistries described in this paper would be a nice piece of work and of interest for a number of researchers in the field of organic and bioorganic chemistry.

I recommend this manuscript for publication in Molecules.